# Elucidating the H^+^ Coupled Zn^2+^ Transport Mechanism of ZIP4; Implications in Acrodermatitis Enteropathica

**DOI:** 10.3390/ijms21030734

**Published:** 2020-01-22

**Authors:** Eitan Hoch, Moshe Levy, Michal Hershfinkel, Israel Sekler

**Affiliations:** Department of Physiology and Cell Biology, Faculty of Health Sciences, Ben-Gurion University of the Negev, Beer-Sheva 84105, Israel; eitanhoch@gmail.com (E.H.); moshe.levy3006@gmail.com (M.L.); hmichal@bgu.ac.il (M.H.)

**Keywords:** ZRT-IRT-like proteins, ZIP, Zinc Transporters, ZnT, zinc transport, ZIP structure function, SLC39A

## Abstract

Cellular Zn^2+^ homeostasis is tightly regulated and primarily mediated by designated Zn^2+^ transport proteins, namely zinc transporters (ZnTs; SLC30) that shuttle Zn^2+^ efflux, and ZRT-IRT-like proteins (ZIPs; SLC39) that mediate Zn^2+^ influx. While the functional determinants of ZnT-mediated Zn^2+^ efflux are elucidated, those of ZIP transporters are lesser understood. Previous work has suggested three distinct molecular mechanisms: (I) HCO3^−^ or (II) H^+^ coupled Zn^2+^ transport, or (III) a pH regulated electrodiffusional mode of transport. Here, using live-cell fluorescent imaging of Zn^2+^ and H^+^, in cells expressing ZIP4, we set out to interrogate its function. Intracellular pH changes or the presence of HCO3^−^ failed to induce Zn^2+^ influx. In contrast, extracellular acidification stimulated ZIP4 dependent Zn^2+^ uptake. Furthermore, Zn^2+^ uptake was coupled to enhanced H^+^ influx in cells expressing ZIP4, thus indicating that ZIP4 is not acting as a pH regulated channel but rather as an H^+^ powered Zn^2+^ co-transporter. We further illustrate how this functional mechanism is affected by genetic variants in *SLC39A4* that in turn lead to Acrodermatitis enteropathica, a rare condition of Zn^2+^ deficiency.

## 1. Introduction

Zn^2+^ is an essential nutrient that plays key roles in a variety of cellular and physiological processes [1]. It is therefore not surprising that Zn^2+^ deficiency, underlined by an inability to acquire nutritional Zn^2+^, has devastating effects. These range from mental disorders, to Immune system dysfunction and growth retardation [2]. The importance of Zn^2+^ to human physiology is further emphasized by a recent finding that approximately 2800 proteins (10% of the human proteome) are potentially Zn^2+^ binding; these include transcription factors, Zn^2+^ finger proteins, and a variety of enzymes [3]. Yet, little is known about the process of Zn^2+^ uptake and how Zn^2+^ ions move across membranes and into cells and organelles.

Two groups of mammalian Zn^2+^ transporters have been identified; SLC39 (ZIP; ZRT-IRT-like protein) mediate Zn^2+^ influx, and SLC30 (ZnT; Zinc transporters) mediate Zn^2+^ efflux [4]. The 10 members of the ZnT family of efflux transporters have been linked to numerous cellular processes that include insulin secretion [5,6] and TNAP activation [7,8]. The functional mechanism of these transporters has been studied in a variety of models, from human cell cultures [9] to plants [10] and bacteria [11]; all indicating a Zn^2+^/H^+^ exchange mechanism. The recently solved structure of a bacterial ZnT orthologue [12] has further enhanced our knowledge on the biochemical and biophysical properties of this group. The 14 members of the ZIP family mediate transport of Zn^2+^ ions into the cytoplasm, either from the extracellular surroundings of the cell, or from intracellular organelles [13]. Members of this group have been linked to various pathologies, such as Ehlers–Danlos syndrome [14,15], and cadmium toxicity [16]. In contrast to ZnTs, our understanding of the mechanisms that govern Zn^2+^ transport by this group is lacking. ZIPs typically have eight transmembrane domains (TMDs), with both N- and C- termini facing the extracytoplasmic side, and a histidine rich domain is found in the cytoplasmic loop between TMDs 3 and 4. The role of this loop is undetermined; however, mutating these residues in the yeast orthologue Zrt1 resulted in mislocalization of the protein, with no effect on Zn^2+^ transport [17]. TMDs 4 and 5 are conserved [18] and highly amphipathic, and thus have been suggested to form a cavity through which ion transport is mediated [4]. Molecular modeling of ZIP4 [19] has recently supported this, and further experimental corroboration comes from IRT1, from *Arabidopsis thaliana*, in which mutating charged residues in TMDs 4 and 5 reduced Fe^2+^ uptake, and reciprocally increased Zn^2+^ uptake [20]. Interestingly, mutating a charged histidine residue in the catalytic core of ZnTs, alters Zn^2+^ vs. Cd^2+^ selectivity [21]. In the current report, we focus on ZIP4 that plays an important role in acquiring nutritional Zn^2+^ [22]. ZIP4 is highly expressed in the small intestines and the embryonic visceral yolk sac, where it primarily localizes to the apical plasma membrane (PM), and undergoes rapid endocytosis, following exposure to Zn^2+^ [23,24]. Under conditions of Zn^2+^ deficiency, ZIP4 is apparently cleaved and a shorter peptide of 37-40 kDa is detected at the plasma membrane [22,24,25], suggesting proteolytic processing regulates ZIP4 expression.

The importance of this transporter is emphasized in individuals with Acrodermatitis enteropathica (AE), a rare human genetic disorder. AE is manifested by several variants of the *SLC39A4* gene [26,27,28,29] that lead to Zn^2+^ deficiency, characterized by skin lesions, growth retardation, immune system dysfunction, and neurological disorders [2,30]. The 3D-structure of BbZIP, a prokaryotic orthologue, was recently identified and several AE-associated variants were mapped onto a ZIP4 model that was based on the solved structure. These variants are clustered around the transmembrane ZIP4 domains and are thought to be critical for ZIP4 homodimerization [31]. ZIP4 has also been signified as a marker for pancreatic cancer [32], leading to elevated Zn^2+^ content in tumor cells, and thus increased cell proliferation and tumor size. Reciprocally, ZIP4 down regulation had a protective effect, limiting tumor growth [33]. Despite the importance of this transporter to human health, the molecular mechanisms by which it mediates Zn^2+^ uptake are unknown.

Previous studies performed on mammalian members of the ZIP family suggested that Zn^2+^ uptake is enhanced either under alkaline conditions or following the addition of HCO3^−^, thus suggesting a Zn^2+^/HCO3^−^ co-transport mechanism. This was suggested for ZIP2 [34], ZIP8 [35] and ZIP14 [36]. On the contrary, studies performed on FrZIP2, a close homologue to ZIP3, obtained from *Takifugu rubripes* (Puffer fish) have shown a reduction of Zn^2+^ uptake following the addition of HCO3^−^ and suggested an increase of Zn^2+^ uptake under acidic pH conditions, suggesting a possible Zn^2+^/H^+^ co-transport mechanism [37]. A recent study mapped the catalytic core of ZIP4 suggesting a pentahedral Zn^2+^ coordination site composed of three histidine and two aspartate residues [38]. Furthermore, the purified and reconstituted ZIP bacterial homologue, ZIPB, acts as a pH regulated slow electrodiffusional channel, and not a transporter, mediating Zn^2+^ transport that is uncoupled from HCO3^−^ or H^+^ transport [39]. Here we monitor cytoplasmic Zn^2+^ and pH changes in HEK293-T cells. Our results indicate that in contrast to the channel-like behavior of the bacterial transporter, the ZIP4-mediated transport of Zn^2+^ and H^+^ is coupled, supporting a Zn^2+^/H^+^ co-transport mode. This suggests that ZIP4 has undergone an evolutionary transformation from a channel to a transporter. We further study how ion transport is affected by two SLC39A4 genetic variants associated with Zn^2+^ deficiency in AE patients.

## 2. Results

### 2.1. Zn^2+^ Transport by ZIP4

Previous studies have shown that ZIP4, as well as other members of the ZIP family, undergoes rapid endocytosis in the presence of extracellular Zn^2+^ [23,24], thus constituting a major experimental challenge in directly monitoring the transport mechanism of ZIP4. Therefore, we initially asked if the rates of transport and endocytosis are sufficiently different to distinguish between. The rate of endocytosis was monitored using the well-established ZIP4 surface-labeling protocol, in which the cells express mZIP4 tagged at its c-terminal with a hemagglutinin (HA) tag facing the extracellular side [23]. Zn^2+^ (50 µM) was added to HEK293-T cells expressing HA-tagged mZIP4 as indicated (Figure 1A). Cells were then washed with ice-cold PBS, and immediately transferred to ice, in order to stop any endocytosis. Subsequently, intact cells were fixed but not permeabilized in PFA and exposed to anti-HA antibodies that thus recognized only the surface bound mZIP4 [23]. Unbound antibodies were extensively washed and level of bound HA, representing surface ZIP4 expression, was determined as a function of exposure time by WB analysis with secondary antibodies to mark the bound anti-HA antibody. Consistent with pervious results [23], no internalization of ZIP4 was observed during the first 2 min of Zn^2+^ exposure (Figure 1A) and a reduction in ZIP4 surface expression was only monitored after 5 min. Our ensuing transport assays were therefore set to a 2-min time interval, following the addition of Zn^2+^, thus allowing accurate monitoring of Zn^2+^ transport, uninterrupted by ZIP4 endocytosis.

Zn^2+^ transport by ZIP4 was monitored in HEK293-T overexpressing mZIP4 or control cells transfected with the empty pcDNA3.1 (empty vector, used as control also in all subsequent experiments) and preloaded with 1 µM Fluozin-3AM, a Zn^2+^ sensitive fluorescent probe, commonly used for monitoring Zn^2+^ transport [21,40]. Cells were perfused in Ringer’s solution containing 50 µM Zn^2+^ and the rate of Zn^2+^ transport was measured and compared to cells transfected with a control vector. Zn^2+^ transport rates mediated by ZIP4 expressing cells were strongly enhanced compared control cells transfected with an empty vector (Figure 1B), indicating that the expression of ZIP4 is linked to enhanced Zn^2+^ transport. To ascertain the increase in Fluozin-3AM fluorescence is triggered by cytoplasmic Zn^2+^, the Zn^2+^ sensitive intracellular chelator TPEN was added at the end of the experiment, following which cytoplasmic fluorescence returned to baseline levels, thus indicating that the fluorescent signal is mediated by changes in cytosolic Zn^2+^. Altogether the results of this part indicate that expression of ZIP4 leads to enhanced Zn^2+^ influx across the PM.

### 2.2. Zn^2+^ Uptake by ZIP4 is pH Dependent

We next sought to determine the mechanism that drives Zn^2+^ transport by ZIP4. In other members of the ZIP family, Zn^2+^ transport was suggested to be coupled to HCO3^−^ [34,35,36], and we therefore sought to determine the effect of HCO3^−^ on Zn^2+^ transport mediated by ZIP4. To address this, HEK293-T cells were transfected with either ZIP4 or an empty control vector, and Zn^2+^ transport was compared in cells perfused with pH7.4 Ringer’s solution containing 50 µM Zn^2+^, in the presence or absence of 20 mM NaHCO3 (Figure 2A). No significant differences were observed, and our results, therefore, did not support a Zn^2+^/HCO3^−^ coupled transport mechanism for ZIP4.

Studies performed on ZIP homologues from other species, such as the bacterial ZIPB [39] and puffer fish FrZIP2 [37], suggest ZIPs act as H^+^ activated Zn^2+^ channels that are independent of HCO3^−^. To determine the effect of pH on Zn^2+^ transport, by ZIP4, we monitored Zn^2+^ transport at the indicated pH values (Figure 2B). Zn^2+^ transport shows strong pH dependency, with a four-fold increase of ZIP4 mediated Zn^2+^ transport rates at pH 5 compared to pH 7.4 (Figure 2C). In contrast, no increase in Zn^2+^ transport was monitored in control cells, expressing an empty vector, at pH5, when compared to pH7.4, indicating that this pH effect is related to ZIP4 activity and not to a non-selective pH dependent change in Zn^2+^ concentrations that may occur by cytosolic acidification [41]. Sensitivity of Zn^2+^ transport to pH was similar to that documented for the bacterial and puffer fish homologues [37,39]. The rates were fitted using a Michaelis–Menten curve and the Hill’s coefficient was 5.1 ± 0.8, indicating a Zn^2+^/H^+^ stoichiometry of 1:5.

To determine if pH controls the apparent affinity or maximal rate of ZIP4 mediated Zn^2+^ transport, we also conducted a Zn^2+^ dose-response analysis at pH 5 and pH 7.4. Utilizing the same experimental design described previously, we monitored rates of Zn^2+^ influx at various Zn^2+^ concentrations ranging from 0–800 µM (Figure 2D). The results were fitted to a Michaelis–Menten equation that suggested that the affinity for Zn^2+^ transport (Km) was pH independent, but the rate of ion transport (Vmax) doubled from 0.5 at pH 7.4 to 1.1 s^−1^ at pH 5, indicating that acidic pH accelerates the turnover rate of Zn^2+^ transport, with no effect on affinity.

Acidic extracellular pH will lead to an intracellular pH drop. The latter can in turn trigger an intracellular Zn^2+^ rise that is independent of ZIP4 activity, by enhancing the dissociation of intracellular bound Zn^2+^ [40,41]. In such conditions, the observed increase in Zn^2+^ accumulation rates may not necessarily be through an extracellular effect on mZIP4, but by a change in cytosolic pH. To address this possibility, we applied the well documented ammonium pre-pulse paradigm [42] that selectively triggers an intracellular, but not extracellular, pH change, in cells preloaded with either 1 µM BCECF-AM (Figure 3A) or 1 µM Fluozin-3AM (Figure 3B). All solutions were Na^+^ free, to prevent the activation of H^+^ efflux via the major cytosolic acid extruder, the Na^+^/H^+^ exchanger. Following intracellular acidification, 50 µM Zn^2+^ were added and both Zn^2+^ and H^+^ transport rates were compared in Zip4 or control cells (Figure 3C). No significant differences were observed indicating the pH activation of Zn^2+^ transport relies solely on extracellular protons. Furthermore, intracellular acidification triggered an inhibition of Zn^2+^ transport by ZIP4 indicating that the reversal of the H^+^ gradient potentially blocked Zn^2+^ transport. Note that a similar result was obtained with FrZIP2, in which Zn^2+^ transport was inhibited following the addition of extracellular HCO3^−^ [37].

### 2.3. Zn^2+^ and H^+^ Transport by ZIP4 Are Coupled

The above results strengthen the hypothesis that extracellular H^+^ ions generate the driving force for ZIP4, suggesting two possible modes of transport for ZIP4: (1) H^+^/Zn^2+^ co-transporter and (2) H^+^ sensitive Zn^2+^ channel. To distinguish between these modes of operation, H^+^ transport was monitored in ZIP4 expressing cells preloaded with 1 µM BCECF-AM. We reasoned that if ZIP4 acts as a channel, no ZIP4 mediated proton transport would be observed.

At pH 7.4, no differences were observed in intracellular pH in the presence of Zn^2+^ (Figure 4A), however at pH 5, a clear rise in cellular acidification was observed in ZIP4 expressing cells, when compared to control cells (Figure 4A,B). This effect is illustrated in Figure 4C that compares the rates of Zn^2+^ transport and pH changes in cells expressing ZIP4, at pH 7.4 (left panel) vs. pH 5 (right panel). Note the strong reciprocity between cytosolic pH acidification and Zn^2+^ influx, which were both enhanced at acidic pH. Altogether our results suggest that ZIP4 mediates H^+^/Zn^2+^ co-transport. Under neutral pH conditions, both Zn^2+^ and H^+^ fluxes were subtle. In an acidic extracellular environment, Zn^2+^ and H^+^ influx rates were strongly increased, supporting an H^+^/Zn^2+^ co-transport mechanism (Figure 4D).

### 2.4. Acrodermatitis Enteropathica Associated Variants Disrupt Zn^2+^ Transport by ZIP4

Genetic variants in *SLC39A4* are linked to Zn^2+^ deficiency in humans [26,27,28,29], but zinc supplementation has not always proven a useful treatment, implying different molecular mechanisms originating from different variants. Several *SLC39A4* coding variants were previously tested and displayed varying levels of expression, as well as varying levels of Zn^2+^ uptake [29]. We focused on two variants of the *SLC39A4* gene, for which surface expression was previously reported [29] and reproduced in our hands (Figure 5A). The P200L variant is a mutation of a residue situated at the cytoplasmic N-terminus domain, and G539R is a residue within the loop connecting TMDs 4 and 5 that form the catalytic site (Figure 5B). Zn^2+^ and H^+^ transport, mediated by ZIP4^P200L^ were no different from those of the wild type ZIP4 (Figure 5C–E), suggesting that the P200L variant is not catalytically linked to the transport activity of ZIP4. Thus, Zn^2+^ deficiency observed in patients harboring this variant may be linked to other processes, e.g., cell surface dynamics of this transporter. Zn^2+^ transport by ZIP4^G539R^, on the other hand, was not activated at acidic pH conditions and maintained basal activity at both pH 7.4 and pH 5 (Figure 5C,D,F), suggesting a role for this residue in pH activation of ZIP4. A prediction for such a role is that H^+^ transport will also be affected. When assayed for H^+^ transport, cells expressing ZIP4^G539R^ demonstrated diminished H^+^ transport at pH 5, supporting a role of this residue in pH activation of the transporter, possibly due to its proximity to the catalytic core formed by TMDs 4 and 5. Thus, the reduction in both Zn^2+^ and H^+^ transport mediated by ZIP4^G539R^ suggests that the substitution of the small non-charged glycine to a larger charged arginine, encountered in AE patients, disrupts the coupling of the H^+^ driving force and Zn^2+^ transport.

## 3. Discussion

### 3.1. ZIP4 Mediates H^+^/Zn^+^ Co-Transport

ZIP4 is a membrane embedded protein, enriched in enterocytes, where it mediates Zn^2+^ uptake. Previous work has shown a regulatory process in which ZIP4 undergoes rapid endocytosis following the addition of extracellular Zn^2+^ [23,24], thus constituting a major challenge in focusing directly on the transport mechanism of ZIP4. To overcome this difficulty, we monitored the timing of ZIP4 removal from the membrane using a HA tagged ZIP4. Tagging may potentially modulate ZIP4 surface expression or activity, however previous studies suggest that its expression pattern and activity are preserved [23]. Endocytosis only begins 2 min following Zn^2+^ exposure. Using live cell imaging, we monitored direct ion transport by ZIP4 during this time thereby addressing the concern that ZIP4 surface expression is changing while its activity is assayed. Notably fluorescence analysis used in this study to monitor Zn^2+^ does not provide an exact measure of transport but of the rate of ion accumulation within the cell. Alternative methods such as inductively coupled plasma mass spectrometry ICP-MS may provide a better quantitative measure for transport rates. Indeed, we have previously used ICP-MS measurements to corroborate the transport mediated by the Yiip transporter [21]. These studies were however carried out on a purified reconstituted Yiip, it is less likely that even overexpression of ZIP4 in HEK293-T cells is sufficiently strong to evoke a zinc signal that can be detected by ICP-MS.

The mechanism by which ZIPs transport Zn^2+^ is by and large unclear. Studies of mammalian ZIPs [34,35,36] have suggested an HCO3^−^ dependent co-transport mechanism, with increased rates of transport either at alkaline pH or following the addition of HCO3^−^. Other studies on the bacterial ZIP homologue, ZIPB [39] indicated that it acts as a pH regulated, electrogenic facilitated diffusion channel, while studies on the puffer fish ZIP homologue, FrZIP2, indicated H^+^/Zn^2+^ co-transport. Our data does not support the involvement of HCO3^−^. The addition of HCO3^−^ did not lead to elevated rates of Zn^2+^ transport, as previously reported also for FrZIP2 [37], and ZIPB [39].

Our results indicate that Zn^2+^ transport by mammalian ZIP4 is mediated by Zn^2+^/H^+^ co-transport, based on the following findings: (1) extracellular acidification triggered ZIP4 dependent Zn^2+^ uptake. (2) In contrast, intracellular acidification—while extracellular pH was kept neutral, inhibited ZIP4 dependent Zn^2+^ uptake. (3) ZIP4 dependent Zn^2+^ uptake, triggered by extracellular acidification, is linked to ZIP4 dependent intracellular acidification. These findings strongly suggest that ZIP4 functions as an H^+^/Zn^2+^ co- transporter.

We suggest that the mammalian ZIP4 transporter underwent an evolutionary progression from the bacterial transporter that acts as a facilitated diffusion channel to an H^+^ coupled Zn^2+^ transporter. What is the physiological advantage of H^+^/Zn^2+^ co-transport as opposed to transport facilitated via diffusion? Facilitated diffusion mediators support transport that is driven by the substrate gradient and allow fast charge movements across the plasma membrane. Indeed, in bacteria, free intracellular Zn^2+^ is vanishingly low [43] and such a mechanism would therefore be optimal for Zn^2+^ uptake. In mammalian cells, on the other hand, Zn^2+^ is an essential rate-limiting nutrient for eukaryotic cells, and Zn^2+^ deficiency is a frequent event with severe physiological consequences. Thus, pathways of Zn^2+^ uptake physiologically favor maximal efficient mechanisms, such as secondary active transport, rather than fast channel uptake systems.

Can an H^+^ coupled transporter support greater Zn^2+^ uptake? Assuming a stoichiometry of 5 H^+^ per Zn^2+^ (see Figure 2C) and based on the Gibbs free energy calculation (see materials and methods), the energy earned from H^+^ transport at pH7.4 would be −0.18674 Kcal/mol, and −17.318 Kcal/mol, at pH 5, thus yielding a Zn^2+^ gradient that is 10^6^-fold higher at pH 5 than that at a neutral pH. Luminal pH, in the proximal small intestines is in the range of 5.5–7. Thus, our suggested mechanism of acidic pH regulation and H^+^ coupled Zn^2+^ co-transport is also supported by the physiology of the gastrointestinal tract and would enable higher nutritional Zn^2+^ absorbance, by utilizing the driving force generated by the H^+^ gradient. Correspondingly, the pH gradient across the plasma membrane of renal tubules would also enable coupling of H^+^ and Zn^2+^, to enhance reabsorption of Zn^2+^. Indeed cells of the proximal tubule abundantly express ZIP4 [27], as well as other members of the ZIP family [44]. Notably, the highly acidic pH dependence of Zn^2+^ transport by ZIP4 differs from the predicted Pka value of the histidine residues at the Zn^2+^ transport site. It could however depend on the Glu/Asp residues at this site. Alternatively, this effect may be triggered allosterically by another site on the ZIP4.

Similar mechanisms of enhanced uptake via either H^+^ or Na^+^ coupling are well documented and abundant in the gastrointestinal tract [45] and renal tubules [46]. The Na^+^/Glucose co-transporter, for instance, couples glucose uptake to the Na^+^ gradient across the plasma membrane to maximize glucose absorption from the intestinal lumen, and reabsorption from the proximal tubules of the nephron [47]. Indeed, in the proximal tubule of the nephron, reabsorption of filtered glucose reaches 100% efficiency. Interestingly, uptake of glucose from the blood stream into erythrocytes and muscle is mediated by facilitated diffusion that maintains constant basal levels of glucose within cells, per the available glucose gradient. Thus, the comparison of Na^+^/glucose co-transport and glucose-facilitated diffusion demonstrates in vivo the importance of an optimal mechanism for maximal absorption. Remarkably, like ZIP4, the renal Na^+^/glucose co-transporter SGLT1 has also been suggested to undergo a regulatory process of endocytosis [48], suggesting these diverse mechanisms did not only develop to support similar energy considerations, but also harbor similar regulatory strategies.

There are, however, risks related to a cellular toxic surge of Zn^2+^ [49,50] following a bolus of Zn^2+^ in the digestive system. This toxic ionic surge can be countered by several documented ZIP4 “safety valve mechanisms”: (a) the slow rate of ZIP4 activity limits Zn^2+^ uptake and thus prevents potential toxicity, (b) the transporter undergoes rapid endocytosis [23,24] that limits the duration of Zn^2+^ uptake, and (c) following uptake of Zn^2+^ from the digestive tract or renal tubules, Zn^2+^ can be rapidly transported, vectorially, into vesicular compartments by the activity of vesicular ZnTs [51] or across the plasma membrane via ZnT1 [52].

### 3.2. Genetic Variants Associated with Zn^2+^ Deficiency in AE Patients Are Linked to Either Catalytic or Non-Catalytic Domains of ZIP4

Genetic variation in ZIP4 is linked to Zn^2+^ deficiency in human subjects [26,27,28,29]. Several of these mutations lead to deletion and frame shift mutations, however, the majority result in single amino acid substitutions. Several of these lead to failure of ZIP4 accumulation at the plasma membrane, possibly due to misfolding that disrupts glycosylation sites and leads to failure in protein localization [29]. Two variants that were previously tested (P200L and G539R) differed from the rest, in that they did accumulate at the PM, yet showed diminished Zn^2+^ uptake over a 15-min time course assay [29].

Our results support the finding that overexpression of ZIP4^P200L^ enables its localization to the PM. However, in our shorter 2 min experiment, Zn^2+^ transport rates were no different from those of the wild type protein, indicating that catalytic Zn^2+^ transport was not impaired by this mutation. A possible explanation to the discrepancy in Zn^2+^ uptake measurements, could be that the P200 residue is found at the extracellular N-terminal domain of ZIP4 that may be sensitive to changes in extracellular Zn^2+^ that regulate the transporter expression [53]. We proposed that this mutated transporter is catalytically active, but undergoes different regulation following Zn^2+^ exposure, which would explain the diminished uptake of Zn^2+^ measured during a 15 min experiment that allows endocytosis, but not observed after 2 min. Thus, the apparent reduction in Zn^2+^ transport following the longer time-course may in fact be related to decreased availability of ZIP4 at the PM. 

In the 15-min experiment, ZIP4^G539R^ also showed diminished Zn^2+^ uptake [53]. However, this mutant also displayed aberration of Zn^2+^ transport in our 2 min transport assay. Interestingly, this residue is located at the loop connecting two transmembrane domains that are part of the putative catalytic domain of ZIP4. Hence, the substitution G539 to a positively charged arginine may directly modulate the catalytic ion transport core and could be attributed to disruption of either Zn^2+^ binding or the conformational change the transporter undergoes.

## 4. Materials and Methods

### 4.1. Cell Cultures

HEK293-T cells (human embryonic kidney cell line) were cultured in Dulbecco’s modified Eagle’s medium (DMEM), supplemented with 10% fetal calf serum (FCS), 1% streptomycin and 1% penicillin. Cells were grown in either 25 cm^2^ or 75 cm^2^ flasks, in a humidified CO_2_ incubator, at 37 °C.

For live-cell imaging and immunocytochemistry experiments, cells were transferred on to glass cover slips, in 60 mm cell culture dishes. For immunoblotting, cells were transferred to 100-mm cell culture dishes.

### 4.2. Plasmid Transfection

Cells were transfected with 0.67 µg of the indicated HA tagged mZIP4 double-stranded plasmid (accession number BC023498, a kind gift from David Eide then in Kansas University [23]) using the well documented CaPO_4_ precipitation protocol in cultures of 40%–60% confluence, 48 h prior to experiment. The original paper [23] shows that the construct faithfully reached the cell plasma membrane and formed a functional transporter that was sensitive to Zn^2+^-dependent endocytosis. The various plasmids used for transfection are described in the following section. Cells transfected with an empty vector—pcDNA3.1—puromycin served as controls in all of the experiments.

Site-directed mutagenesis was performed using the QuikChange site-directed mutagenesis kit (Stratagene, San Diego, CA, USA) according to the following protocol (Table 1 and Table 2). 

The following primers were designed using the primer design tool, on the University of Washington server (http://depts.washington.edu/bakerpg/webtools/PD.html) and manufactured by SIGMA (St. Louis, MO, USA).

ZIP4^P200L^: CCAAGGCCTG**CTT**AGCCCTCAGTA.

ZIP4^G539R^: CCATGAACTC**CGC**GACTTCGCTGCTCTG.

### 4.3. Immunoblot Analysis

Cells were extracted using 200 µL of boiling denaturative lysis buffer (1% SDS, 10 mM Tris-HCl, pH 8)/100-mm plate and transferred to ice. A protease inhibitor mixture (Boehringer Complete protease inhibitor mixture; Roche Applied Science) was added to the lysates, and protein concentrations were determined using the modified Lowry procedure [54]. SDS-PAGE and immunoblot analyses were performed, using anti-actin and anti-HA antibodies at dilutions of 1:40,000, and 1:2000 respectively. Secondary anti-mouse and anti-rabbit antibodies (Jackson Immunoesearch, West Grove, PA, USA) were used at dilutions of 1:20,000 and 1:40,000, respectively. Densitometry analysis of expression level was performed using EZQuant-Gel software (EZQuant, Hertzliya Pituach, Israel) as previously described [55].

The detection of membrane embedded ZIP4 the protocol described by Kim et al. [23] was used. Cells overexpressed mZIP4 tagged at its c-terminal with a HA tag facing the extracellular side. Cells were incubated with 50 µM Zn^2+^ for 0–60 min, as indicated. Zn^2+^ uptake and endocytosis were terminated by transferring the cells to ice and subsequently washing the cells with ice-cold PBS. Following this step, cells were fixed using 4% PFA in 0.1 M PBS, without permeabilization of the cells. The HA tag at the extracellular c-terminus of mZIP4 enables us to label only the cell surface associated mZIP4. Cells were then washed in PBS to remove residual PFA, and following the removal of PBS, cells were incubated with 1 µg/µL anti-HA antibody, for 30 min at room temperature. The anti-HA antibody was recovered from the non-permeabilized intact cells and therefore marked the surface expression of mZIP4 using the previously demonstrated method [23]. Cells were then washed five times with PBS, to remove residual unbound antibodies and immediately exposed to boiling denaturative lysis buffer for western blotting. Following SDS-PAGE and immunoblotting, membranes were incubated with secondary anti-mouse antibody.

### 4.4. Live Cell Fluorescent Imaging

The imaging system consisted of an Axiovert 100 inverted microscope (Zeiss, Oberkochen, Germany), Polychrome II monochromator (TILL Photonics, Planegg, Germany), and a SensiCam cooled charge-coupled device (PCO, Kelheim, Germany). Fluorescent imaging measurements were acquired with the Imaging Workbench 6 software (Axon Instruments, Foster City, CA, USA) and analyzed using Microsoft Excel, Kaleidagraph and Matlab.

Cytoplasmic Zn^2+^ transport was determined in cells loaded with 1 µM Fluozin-3AM. To verify that the fluorescence changes were related to intracellular ions, the cell-permeable heavy metal chelator *N*,*N*,*N*’,*N*’-tetrakis-(2-pyridylmethyl)-ethylenediamine (TPEN; 20 µM) was used.

Cytoplasmic pH changes, indicative of H^+^ transport, were determined in cells loaded with 1 µM BCECF-AM, a pH sensitive dye. Values of intracellular pH were calibrated using high K^+^ Ringer’s solution set to pH values of 6–8 in the presence of Nigericin [56].

Intracellular acidification was triggered using the ammonium prepulse paradigm [42]. Cells were superfused with Ringer’s solution containing NH4Cl (30 mM, replacing the equivalent 30 mM NaCl), which was subsequently replaced with NH4Cl-free solution, thus triggering intracellular acidification.

### 4.5. Statistics

Data analysis was performed using the SPSS software (version 14.0; SPSS Inc., Chicago, IL, USA). All results shown are the means ± S.E. of at least three individual experiments (*n* ≥ 3) in each several coverslips and several regions of interests (ROI) within coverslip were independently monitored and included in the analysis. Two tailed *t*-test *p* values of ≤ 0.05 were considered significant following Levene’s test for equality of variances. Significance of the results # is *p* ≤ 0.05, between the indicated bars.

## Figures and Tables

**Figure 1 ijms-21-00734-f001:**
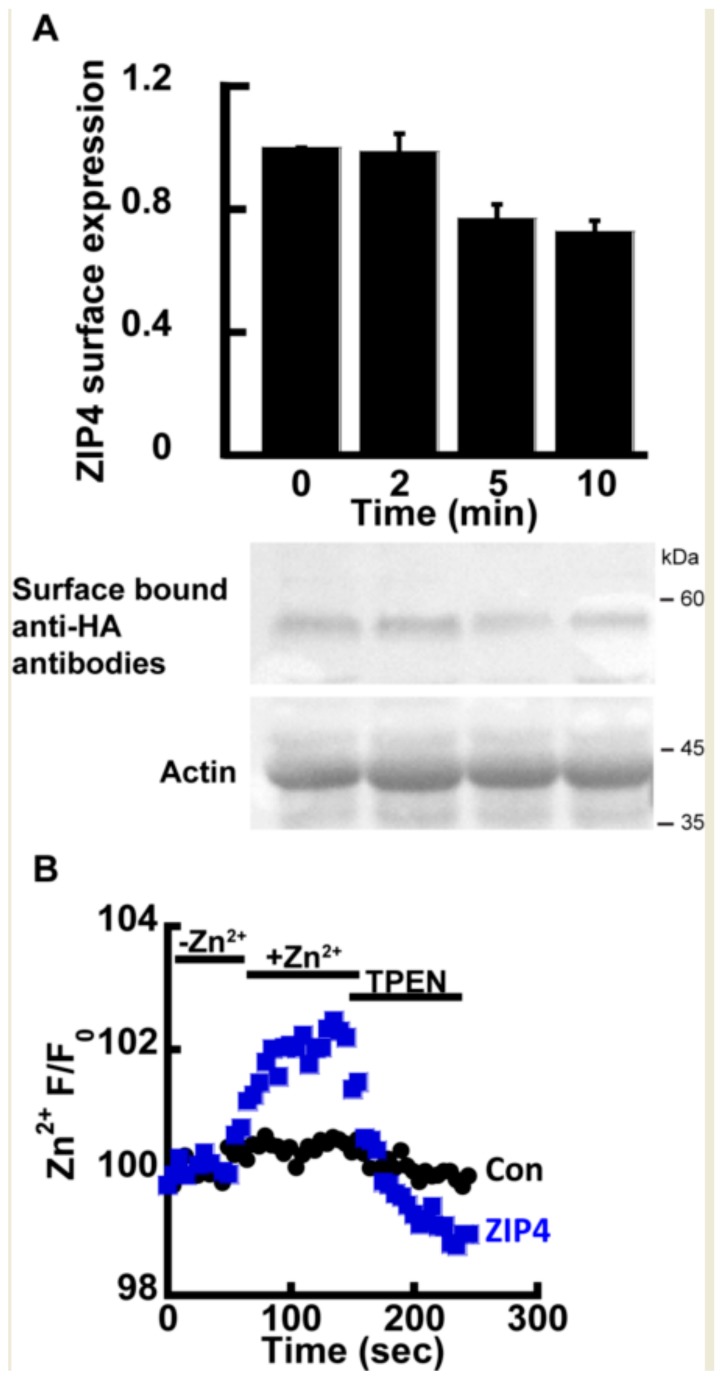
ZIP4 Zn^2+^ transport assay. (**A**) Immunoblot analysis (Lower panel) and normalized surface expression by densitometry (upper panel) of HEK293-T cells transfected with hemagglutinin (HA)-tagged mZIP4 and exposed to 20 µM Zn^2+^ for the indicated times *N* ≥ 5. (**B**) Experimental assay used to monitor Zn^2+^ uptake in HEK293-T cells transfected with an empty control vector (black) or mZIP4 (blue) *N* ≥ 5. Cells were loaded with 1 µM Fluozin-3AM and subjected to 50 µM Zn^2+^ and 20 µM TPEN, as indicated.

**Figure 2 ijms-21-00734-f002:**
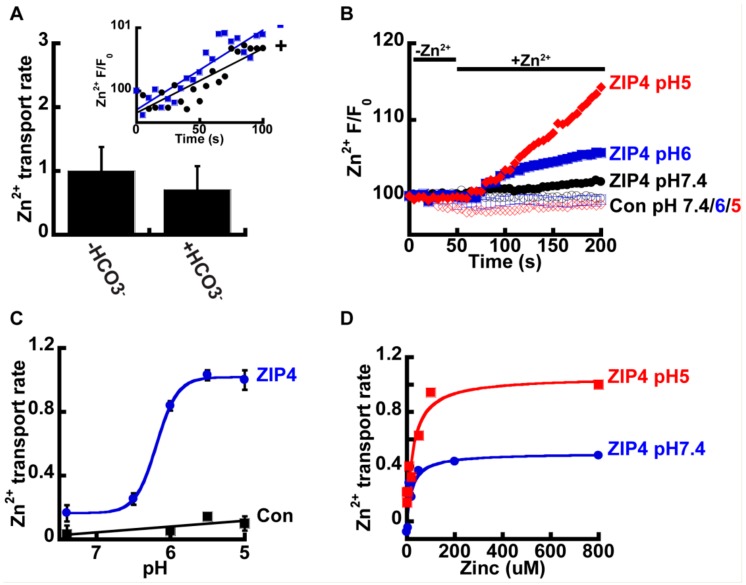
ZIP4-mediated Zn^2+^ transport is pH dependent. (**A**) HCO3^−^ has no effect on Zn^2+^ uptake. HEK293-T cells transfected with ZIP4 were loaded with 1 µM Fluozin-3AM and loaded with 50 µM Zn^2+^ in HEPES buffered Ringer’s solution (blue, in inset) or 20 mM NaHCO3^−^ buffered Ringer’s solution (black, in inset) *N* ≥ 3. Normalized Zn^2+^ uptake rates in the presence or absence of HCO3^−^ and is shown in the bar graph. (**B**) Representative traces of Zn^2+^ uptake, in HEK293-T cells, transfected with an empty control vector (empty symbols) or mZIP4 (full symbols) that were applied in Ringer’s solution at different pH levels, as indicated. (**C**) Normalized rates of Zn^2+^ uptake (compared to the rate of transport by mZIP4 at pH 5) by cells expressing either mZIP4 or a control vector *N* ≥ 5. Curve is a Michaelis Menten fit. (**D**) Rates of Zn^2+^ uptake, in HEK293-T cells expressing ZIP4, at pH 7.4 (blue) and pH 5 (red), at the indicated zn^2+^ concentrations (0–800 µM). Curve is a Michaelis Menten fit *N* ≥ 4.

**Figure 3 ijms-21-00734-f003:**
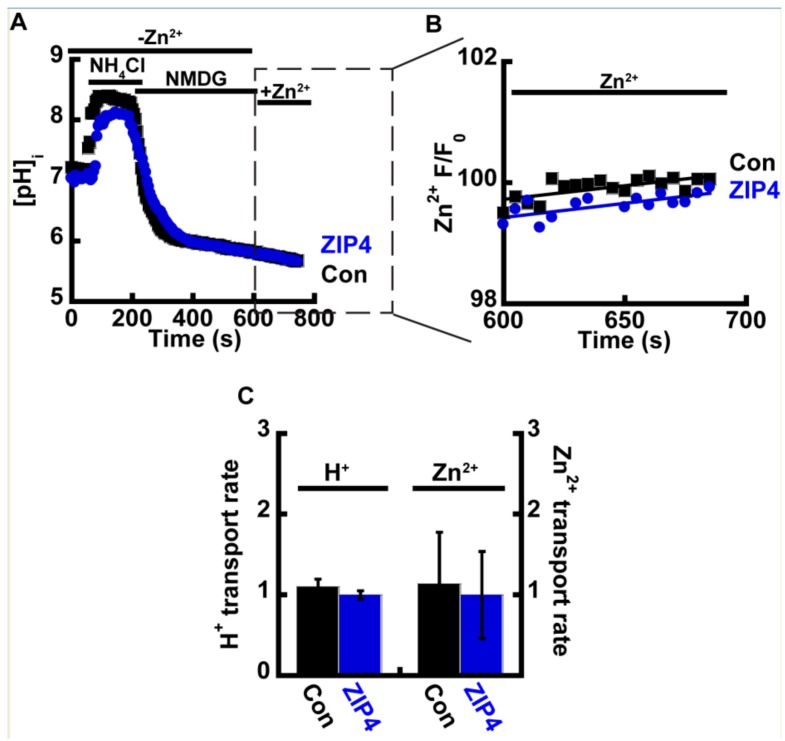
Intracellular acidification limits Zn^2+^ uptake. (**A**) Representative traces of HEK293-T cells transfected with ZIP4 (blue) or an empty control vector (black), loaded with 1 µM BCECF-AM, and subjected to ammonium pre-pulse paradigm to monitor cytoplasmic pH changes without extracellular acidification. (**B**) Corresponding Zn^2+^ traces in cells loaded with 1 µM Fluozin-3AM. Note that traces are presented only upon the addition of Zn^2+^, following acidification. (**C**) Normalized H^+^ and Zn^2+^ transport rates N ≥ 4.

**Figure 4 ijms-21-00734-f004:**
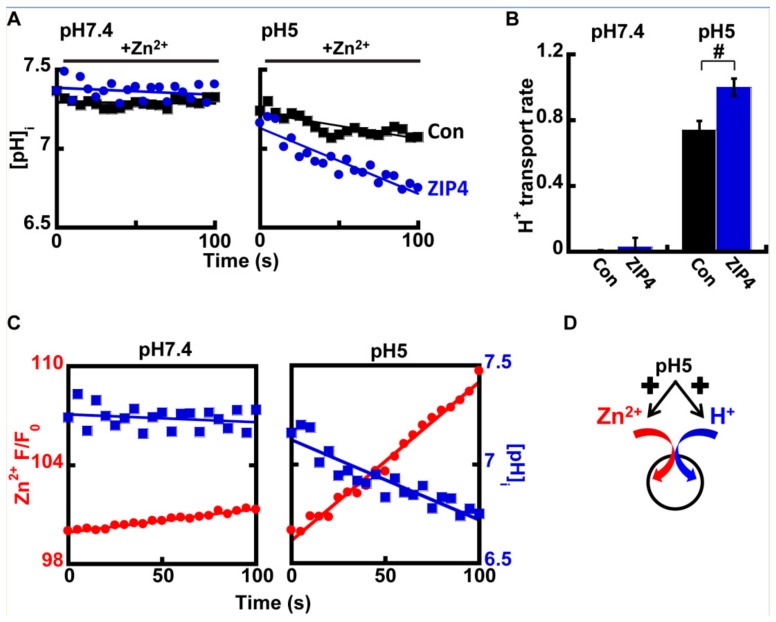
ZIP4 mediates H^+^ coupled Zn^2+^ transport. (**A**) Representative traces of HEK293-T cells transfected with ZIP4 (blue) or a control vector (black), loaded with 1 µM BCECF-AM, and monitored for cytoplasmic pH changes at pH 7.4 (left) or pH 5 (right), following the addition of 50 µM Zn^2+^. (**B**) Normalized H^+^ transport rates recorded from control (black) or ZIP4 (blue) expressing cells, in the presence of 50 µM Zn^2+^ in the extracellular solution *N* ≥ 5. (**C**) Representative traces of Zn^2+^ (red) and H^+^ transport (blue), recorded with Fluozin-3AM and BCECF-AM accordingly. Note that H^+^ uptake is parallel to Zn^2+^ uptake. (**D**) Illustration of the suggested mechanisms of ZIP4. In (B), # is *p* ≤ 0.05 between control cells and cells expressing ZIP4 at pH 5.

**Figure 5 ijms-21-00734-f005:**
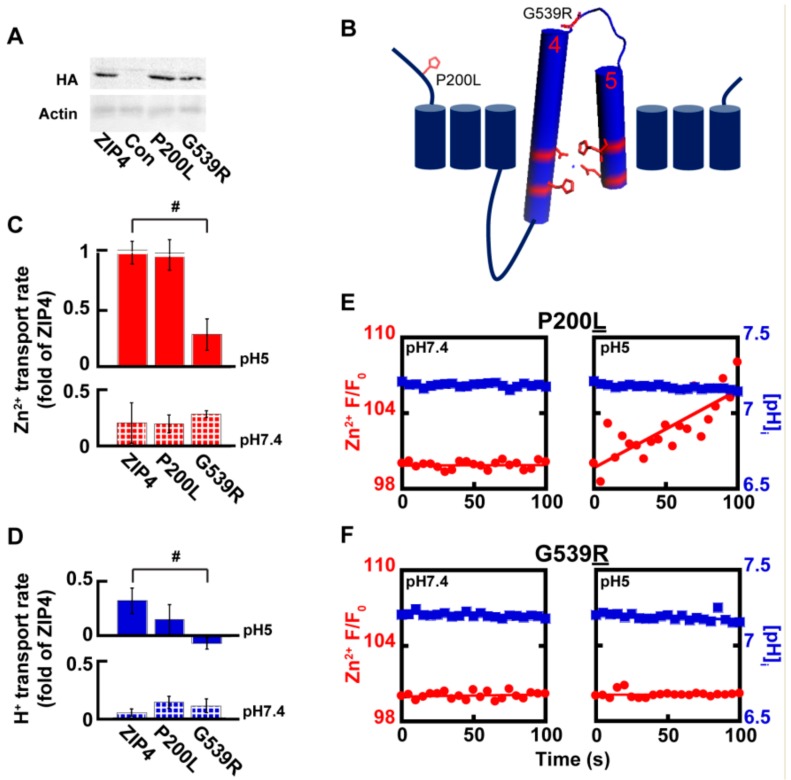
Genetic variations in ZIP4 affect catalytic and non-catalytic domains. (**A**) Immunoblot analysis of Acrodermatitis enteropathica (AE)-associated ZIP4 constructs, as indicated, from HEK293-T cell lysates. (**B**) Membrane orientation of ZIP4 illustrates the position of AE-associated variants P200L and G539R. (**C**) Normalized rates of Zn^2+^ uptake at pH7.4 (bottom panel—striped) and pH5 (top panel—filled), mediated by AE-associated ZIP4 mutants, in HEK293-T cells loaded with Fluozin-3AM *N* ≥ 5. (**D**) Normalized rates of H^+^ uptake at pH 7.4 (bottom panel—striped) and pH5 (top panel—filled), mediated by AE-associated ZIP4 mutants, in HEK293-T cells loaded with BCECF-AM. (**E**–**F**) Representative traces of Zn^2+^ (red) and H+ uptake (blue) recorded in HEK293-T cells transfected with ZIP4^P200L^ (**E**) and ZIP4^G539R^ (**F**) *n* ≥ 3. In (**C**) and (**D**), # is *p* ≤ 0.05 between cells expressing ZIP4 and G539R mutant at pH5.

**Table 1 ijms-21-00734-t001:** Site directed mutagenesis reaction components.

Component	Amount
dsDNA template (100 ng)	0.333 µL
Primer F (0.3 µM)	0.75 µL
Primer R (0.3 µM)	0.75 µL
dNTPs	1.5 µL
Enzyme	0.5 µL
ddH_2_0 (a total of 25 µL)	21.167 µL
Total volume	25 µL

**Table 2 ijms-21-00734-t002:** Site directed mutagenesis PCR.

Cycles	Step	Temperature	Time (min:s)
1	Initial denaturation	95 °C	5:00
20	Denaturation	98 °C	0:20
	Annealing	78 °C	0:20
	Elongation	72 °C	0:30/1 kb template
1	Final elongation	78 °C	5:00

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
