# Peer review of "Elucidating the H+ Coupled Zn2+ Transport Mechanism of ZIP4; Implications in Acrodermatitis Enteropathica"

_ijms, 2020, doi:10.3390/ijms21030734_

Round 1
Reviewer 1 Report
This paper is a well written and interesting manuscript that clearly explains the co-transport ability of ZIP4, regarding H+ and Zinc. This is an important piece of work as ZIP transporters are increasing in importance in health and disease yet we still do not know whether they are even channels or transporters. The fact that they are evidenced here to co-transport zinc and H+ offers a key breakthrough to help the whole broader scientific community.
I found no problems in the paper except for a few very minor issues listed below.
last line before results 'SLC39A4', line before that 'from'
section 2.1 , line 12, 'previous'
page 8, 8 lines from bottom, 'coupling of H+'
page 8, 5 lines from bottom, 'via'
perhaps the authors should mention in discussion about mutant G539R the potential affect that having a positively charged residue instead of the G would have on transport.
Author Response
Thanks pointing out the typographical errors. These have been corrected.
The potential impact of G539R on transport has been added to the discussion section.
Reviewer 2 Report
This is a very well conceived and performed study, the results are convincing and the discussion is thorough. Still, I have two concerns that need to be addressed in the revised manuscript. The first regards the use of FluoZin-3. The coordination chemistry of this sensor has been recently revisited and serious issues raised about the interpretation of its fluorescence under intracellular conditions (Marszalek et al. Inorg. Chem. 57, 9826−9838, 2018). I therefore recommend confirming the Zn2+ influx by a sensor-independent method, e.g. ICP-MS. The second is about the pH independence of Zn2+ affinity to ZIP4. For each His in the binding site there should be a one log unit of affinity lost at pH 5 vs. pH 7.4. Therefore, either the site is composed solely from Asp/Glu, or the pH difference is not sensed at the site. This issue requires a comment.
There are minor typing errors occasionally, in particular in chemical formulae. The amino acid names should not be capitalized.
Author Response
Thanks for raising the important issue about the use of fluorescent sensors to monitor intracellular zinc and suggesting the use of an alternative method (i.e. ICP-MS) to confirm our findings. Indeed we have previously performed ICP-MS measurements to corroborate the transport mediated by the Yiip transporter (Hoch et al PNAS 2012). Note that these studies were carried on a purified reconstituted Yiip. It is unlikely that the expression of ZIP in HEK cells is sufficiently strong to evoke a zinc signal detected by ICP-MS. We now relate to this issue in the discussion and added the relevant citation to the text.
Thank you for raising the pH dependence issue of ZIP transport. We fully agree that it does not agree with the predicted Pka value of the His residues at the Zn2+ transport site, and could therefore depend on the Glu/Asp residues at the transport site. Alternatively, this effect may be triggered allosterically by another site. We have included this in the revised discussion.
We revised and corrected all the other minor issues raised by this reviewer.
Reviewer 3 Report
In this study, Hoch et al., investigate the molecular mechanisms by which ZIP4 is realizing zinc uptake into cells. Although this is a very interesting topic and the study also shows interesting data, there is a lack of details in the manuscript. It is necessary to significantly reduce the superficiality of the description of the results and experimental procedures.
Major points:
A major problem of the study is the use of a mouse Zip4 transporter expressed in a human cell line with a tag (please explicitly mention whether the tag was N- or C-terminal). Technically, it would have been no problem to use either human Zip4 or a mouse cell line. The authors need to explain the rationale behind the unfortunate combination and provide evidence that mouse and human Zip4 behave the same way in HEK cells. In Figure 1, please provide the molecular weight in the western blot image. Please add: How was the quantification of WB bands performed. Please include in all figures, the number (n=?) used for the statistical analysis. Using a t-test is not correct in most cases. For example, the results in Figure 1A need to be evaluated by a one-way ANOVA followed by (in case of significance) a Tukey or Bonferroni posthoc test for pairwise comparison (the same applies to Figure 5C and D).The results in Figure 4B are evaluated regarding two factors, pH and Ctrl vs Zip4. Thus, a two-way ANOVA with Post-hoc analysis needs to be performed. In the materials and methods section, please provide detailed information regarding the control vector used. The text says “The various plasmids used for transfection are described in the following section“, but the description is missing. Please comment on the stoichiometry of Fluozin-3AM and zinc. How much of the added 50uM zinc can be detected by 1uM Fluozin? Why were 20uM TPEN selected? Why does the fluorescence intensity return to baseline and not 0, as TPEN should scavenge all available free zinc in cells. Please provide a rationale for using 20uM NaCHO3. In what way is this reflecting the physiological conditions? Using 800uM zinc, even for a short time (Figure 2D), have the authors considered (excluded) effects on cell health? Does this treatment change the osmolarity considerably? Have the authors considered using, for example, 50uM zinc and 450uM mannitol to control for the influence of changed osmolarity? Although the authors mentioned several times that they use well-established protocols, this does not mean that they do not have to a) perform the necessary controls and b) show that the protocols work in their hands. For example, how do they know that the “ammonium pre-pulse paradigm [41] that selectively triggers an intracellular, but not extracellular, pH change“ really only changed intracellular pH in their setup? The necessary data should be shown (e.g., supplementary information).
Minor points:
An information that I would include in the abstract is the type of cell that has been used for the experiments. the sentence: “Interestingly, mutating a charged Histidine residue in the catalytic core of ZnTs, alters Zn2+ vs. Cd2+ selectivity [20]” is not well connected to the previous paragraph on Zips and should be moved further to the beginning, where ZnTs are discussed. Please briefly discuss the presence of endogenous Zip4 in HEK cells and whether this may have influence on the results. Please change uM to μM The authors introduced mutations reported in humans into mouse Zip4. Can the authors please comment whether the sequence of human ZIP4 and mouse ZIP4 is identical at these sites?Author Response
Thank you for relating to the species issue. The major purpose of this work is to determine biochemical and molecular aspects of ZIP4 function, and not to examine signaling, physiological or whole organism effects, for which a different experimental system would be needed. Specifically, the HEK293-T cells are well suited for our experimental paradigms since they do not exhibit high capacity of endogenous zinc transport. Overexpression of the mZIP4 transporter is required to monitor any zinc transport in these cells, and thus provides a good model to study the functional mechanism of this class of transporters that is yet poorly understood. Indeed, HEK293-T cells are routinely used to study zinc transport.
The information regarding the HA tag and WB analysis of surface expression (Fig. 1) follows a protocol by Eide (Wang et al., JBC 2004) and is now better explained in the text.
Regarding the statistical analysis. While the bar graphs represent results of several experimental conditions, the comparison was always made between an independent variable and its specific control. Thus the T-test is acceptable for this type of analysis. We have now added marks to explain which conditions were compared in each experiment.
The control vector and the plasmids are now described in the Methods. Thank you for raising this issue.
To address the several concerns made by this reviewer regarding reagents and concentrations used in our study (i.e. fluozin-3AM, TPEN and NaCHO3) we note that we have applied paradigms that have been previously used to study functional aspects of zinc transport. Indeed some of these concentrations may not be reflecting physiological conditions but have an important role for the biophysical analysis of the transporter activity. We have provided references to previous papers using these tools in similar manner. The intracellular zinc levels are determined by a complex interaction between zinc buffering proteins, the fluorescent dye and the chelator. Not only their affinity but also association/dissociation constants are important. Therefore in our experiments, we rely on the initial rate of fluorescence change which reduces at least some of these hurdles and allows us to compare rates of transport mediated by the transporter. The fact that we see a clear difference between initial transport rates of cells expressing ZIP4 and control cells indicate that these rates reflect genuine transport by the ZIP. It is likely that if the incubation time with TPEN would have been longer or higher concertations would be used, it is conceivable that the drop in fluorescence would have neared "zero values". However this is not relevant for our analysis of the ZIP-dependent transport. In addition, the overall change in osmolality when we applied 800 uM Zn is negligible considering the ~ 200 mM total osmolarity of the solution (the Zn change is less than 0.5%) and therefore we not osmotically compensate for the addition of Zn2+. Importantly, our dose response experiments were limited to two minutes; any impact of high zinc concentrations on cellular health is negligible under these conditions, and irrelevant to the mechanistical characterization of the transporter. Indeed saturation of the transport is observed at this concentration.
Thank you for raising the issue of the ammonium prepulse experiments. It is a very well established procedure to selectivity change the pHi and hundreds of papers used it to faithfully modulate pHi. The solutions used for the extracellular perfusion are all maintained at pH 7.4 in this protocol, thus any concern regarding extracellular pH is irrelevant in this well-established paradigm. We also used it extensively, in previous studies, to monitor the pH dependence of intracellular sites of transporters (see Ohana et al JBC 2004, Ohana et al JBC 2009, Hoch et al PNAS 2012).
Textual changes were made to address the reviewer’s concerns regarding the description of the plasmids used in the study, and the description of WB quantification.
The mutations of the mZIP4 have all been aligned with the human mutations, and this is now clearly mentioned in the text. We thank the reviewer for this comment.
Reviewer 4 Report
The authors present an interesting and novel work characterizing the Zn2+ and proton transport activity of ZIP4 transfected HEK cells. The work has strong merit but addressing a few questions would further strengthen the results.
1.) How was the zinc concentration of 50 uM chosen? Typically, it would be expected that multiple zinc concentrations (e.g. 1 uM, 10 uM, 100 uM etc.) would be tried. In figure 4 for example it seems that employing multiple zinc concentrations could show also a zinc-concentration dependence of proton transport rate.
2.) How was the 20 uM HCO3- concentration chosen?
3.) Was TPEN used as control only at pH 7.4? And the methods indicate that TPEN was applied at 20 uM whereas Figure 1 indicates TPEN was used at 50 uM.
Overall a very fine scientific work, and addressing these concerns would make it even stronger.
Author Response
Thank you for raising these important questions. The Zn2+ concentration is based on the initial dose response performed on the ZIP4 transport, and is relevant to the concentration of zinc found in the digestive tract where ZIP4 is expressed. Note that proton transport dose response is more challenging because of the high background of proton leaks at acidic pH. Indeed further analysis of this issue is of interest but due to the challenges out of the scope of the current paper. This TPEN concentration were chosen by their ability to chelate Zn2+ at the timecourse used for our analysis. Higher TPEN concentrations can also chelate Ca2+ and complicate the results of the experiments. TPEN was indeed used at 20μM, and we thank the reviewer was catching this typographical error in the figure. The HCO3- concentration used was 20mM, we apologize for the mistake. This concentration is related to its physiological concentration based on the published literature discussing ZIP4 endocytosis and ZIP8/ZIP12/ZIP14 HCO3- transport by the Lab of David Eide. The relevant papers are cited in our manuscript.
Round 2
Reviewer 3 Report
The authors have successfully addressed some of the criticism, however, major issue remain that need to be resolved. It is especially frustrating that the authors do not respond to all the points raised in the first review process.
Major points:
A) The use of a mouse Zip4 transporter expressed in a human cell line with a tag (please explicitly mention whether the tag was N- or C-terminal) needs to be mentioned as limitation in the Discussion section.
B) In Figure 1, please provide the molecular weight in the western blot image. Please add: How was the quantification of WB bands performed. Please include in all figures, the number (n=?) used for the statistical analysis.
C) Using a t-test is not correct in most cases. The authors explain:
"While the bar graphs represent results of several experimental conditions, the comparison was always made between an independent variable and its specific control. Thus the T-test is acceptable for this type of analysis."
Is is simply wrong. If these are independent experiments, they cannot be shown in one single graph. Figure 1A, 4B, 5C,D cannot be evaluated by t-tests.
D) In the materials and methods section, please provide detailed information regarding the control vector used. I still do not see this information in the updated version.
E) Even if the authors use "very well established procedure to selectivity change the pHi and hundreds of papers used it to faithfully modulate pHi." This does not mean that the authors do not have to perform the proper control experiments or have to show evidence that the protocol really worked in their hand. This is unscientific. The authors need to provide evidence that the protocol worked.
Author Response
We were very pleased to see that reviewers 1,2 and 4 were fully satisfied with the revisions that we made . We also address as detailed below the remaining issues raised by reviewer 3:
Thank you for raising the tag issue. The tag that has been used is facing the extracellular side, and thus does not pose any limitation to the use. Indeed, in a previous study this tagged Zip4 was thoroughly characterized and its activity and expression were intact (Kim et al. JBC 2004). As this procedure monitors only the surface bound antibody following lysis of the cells by sample buffer, the molecular weight does not represent Zip4, but the bound antibody. As such, the molecular weight is not presented even in the original paper, and will be therefore misleading and confusing. We marked it instead with the label "Surface bound HA antibodies" to make this point clearer. The use of t Test when two groups are compared is indeed justified, and it is very common to put the relevant experiments on the same graph, which helps the reader see the full picture, while specifically marking significance only between those groups that are compared. For example, see figure 4 in the MS by Amendola et al. Nature 2019 (https://www.nature.com/articles/s41586-019-1832-9). We do not see a reason to separate the graphs and present something that is much less common in the scientific community and will again only confuse the readers. Please note that we in fact show the ammonium prepulse. The MS (Fig. 3) shows the intracellular pH throughout the ammonium prepulse experiment, such that it is clear that in our hands this procedure indeed acidifies the intracellular pH. The extracellular pH is the pH of the perfusing solution that is calibrated to 7.4. The left panel shows the corresponding Zn2+ response, at the time interval that is marked with the dashed line to make it clearer. We have related to the tagged protein issue in the discussion. Indeed tagging may sometimes impair protein activity but this tagged protein was previously extensively studied by the group of Eide and shown to be functional. Following the reviewer request we now relate to this issue in the discussion. We have now added the n for number of experiments in all the relevant figure legends.Round 3
Reviewer 3 Report
The authors are unwilling to address my concerns. The study should be rejected. It uses the wrong statistical tests and lacks appropriate description of the material and methods.
Author Response
Thank you for the remaks, we have revised the MS accordingly. Specifically:
1. Molecular markers were added in Fig 1A.
2. The plasmid was not generated by us, but was generated by Prof. Eide who also published a detailed MS using the same plasmid and the methods. We have modified the methods section to describe the procedure of using it, which we have done in the MS. We have clearly cited the original paper.
3. We have now added the explanation regarding the statistical comparison in each relevant figure legend.